# Synergistic Antibiofilm Effects of Ultrasound and Phenyllactic Acid against *Staphylococcus aureus* and *Salmonella enteritidis*

**DOI:** 10.3390/foods10092171

**Published:** 2021-09-13

**Authors:** Jiaojiao Zhang, Debao Wang, Jinyue Sun, Zhilan Sun, Fang Liu, Lihui Du, Daoying Wang

**Affiliations:** 1School of Food and Biological Engineering, Jiangsu University, Zhenjiang 212013, China; jjz1234560722@163.com; 2Key Lab of Food Quality and Safety of Jiangsu Province-State Key Laboratory Breeding Base, Nanjing 210014, China; Debaowang_2021@163.com (D.W.); jinyue952021@163.com (J.S.); sunzhilan@jaas.ac.cn (Z.S.); daoyingwang@yahoo.com (D.W.); 3Key Laboratory of Grains and Oils Quality Control and Processing, Collaborative Innovation Center for Modern Grain Circulation and Safety, College of Food Science and Engineering, Nanjing University of Finance and Economics, Nanjing 210023, China

**Keywords:** *Staphylococcus*, *Salmonella*, biofilm, dehydrogenase, polysaccharides

## Abstract

This study evaluated the effect of the combination of ultrasound and phenyllactic acid (PLA) on inactivating *Staphylococcus aureus* and *Salmonella enteritidis* biofilm cells and determined the possible antibiofilm mechanism. *S. aureus* and *S. enteritidis* biofilm cells were separately treated with ultrasound (US, 270 W), phenyllactic acid (PLA, 0.5% and 1%), and their combination (US + 0.5% PLA, and US + 1% PLA) for 5, 10, 20, 30, and 60 min. Biofilm inactivation, polysaccharide, and respiratory chain dehydrogenase assays were conducted. US and PLA had a synergistic effect on inactivating bacterial cells in *S. aureus* and *S. enteritidis* biofilms. The combination of US and PLA significantly decreased the contents of soluble and insoluble polysaccharides and the activity of respiratory chain dehydrogenase in the biofilm cells compared to the single treatment. Confocal laser scanning microscopy, scanning electron microscopy, and intracellular adenosine-triphosphate (ATP) analyses indicated that the combination of US and PLA seriously destroyed the cell membrane integrity of the *S. aureus* and *S. enteritidis* biofilms and caused the leakage of intracellular ATP. These findings demonstrated the synergistic antibiofilm effect of US combined with PLA and offered a research basis for its application in the food industry.

## 1. Introduction

Food safety has been a serious threat to people’s health given the frequent outbreak of foodborne diseases and has attracted increasing attention worldwide [1]. Foodborne pathogens, such as *Staphylococcus aureus* and *Salmonella enteritidis*, are of worldwide concern due to their latent hazards to food security and public hygiene [2]. The United States recalled 31 foods from the food market due to *S. enteritidis* (https://www.fda.gov/Food/Recalls, accessed on 24 January 2019). In China, 25% of bacterial poisoning incidents were caused by *S. aureus* [3]. *S. aureus* and *S. enteritidis* can form biofilms on solid surfaces [4,5]. A biofilm comprises strongly enclosed microorganisms that cling to a surface and/or to each other, generating complex actionable circumstances [6,7,8]. Biofilms are found on food contact structures, such as band carriers, vessels, and pipes [9]. The formation of biofilms provides protection for foodborne pathogens and ensures that bacterial cells can adapt to unfavorable growth conditions. The elimination of pathogen biofilms is the key to ensure food safety.

Ultrasonic sterilization (US) is an effective method that can inactivate bacteria and is more suitable for liquid foods [10]. This method can inactivate common foodborne pathogens, such as *Listeria* and *Escherichia coli*, in dairy products and can ensure the preservation of their nutritional composition [11]. In addition, US is environmentally friendly and can be used to clean the surfaces of machines in the food industry [12,13]. However, a single ultrasonic treatment can only inactivate *Cronobacter sakazakii* biofilm cells by 0.2 log CFU/cm^2^, indicating the limited effect of this process on removing biofilm cells [14]. Selan et al. [15] found that when ultrasound and antibiotics work together, on the basis of ensuring the same bactericidal effect, the presence of ultrasound can reduce the dose of antibiotic ampicillin. Alenyorege et al. [16] used the combination of ultrasound and sodium hypochlorite and found that it significantly reduced the number of *Listeria innocua*. Duarte et al. [17] found that the combined ultrasound and dichloroisocyanurate had significantly improved the antibacterial effect. In addition, ultrasound can improve the antibacterial effect of oregano essential oil and organic acids [18,19]. Phenyllactic acid (PLA) has a broad spectrum of effective antibacterial activity against many Gram-positive bacterial species, including *S. aureus* and *Listeria monocytogenes*, as well as Gram-negative bacterial species, including *S. enteritidis* and *E. coli*. Liu et al. [20] reported that PLA decreased the biofilm formation of *Enterococcus faecalis*. Liu et al. [21] found that PLA significantly inactivated *L. monocytogenes* biofilm cells and that the bactericidal effect was enhanced with increasing PLA concentration.

The combination of ultrasound and PLA to inactivate foodborne pathogens in biofilms has not been reported. The aim of this study was to evaluate the combined effect of ultrasound and PLA on inactivating the biofilm cells of *S. aureus* and *S. enteritidis* and to explore the possible antibiofilm mechanism. Results provide a new method to control *S. aureus* and *S. enteritidis* biofilms in the food industry and other fields.

## 2. Materials and Methods

### 2.1. Bacterial Cultivation

*S. aureus* and *S. enteritidis* strains were stored at −80 °C and cultured in Brain Heart Infusion broth (BHI, Qingdao Rishui Bio·Technologies Co., Ltd., Qingdao, China) at 37 °C for 24 h. The strains were streaked on BHI agar and cultivated at 37 °C for 24 h. Individual colonies were inoculated in tubes containing 5 mL of BHI broth and transferred to fresh BHI broth every 24 h, replicated three times. At the final culture stage, the bacterial count was confirmed by plating on BHI agar with a 1:10 series dilution.

### 2.2. Treatment Parameters

The concentrations of PLA used in the experiment were 0.5% and 1% (*w*/*v*). For the PLA treatments, mature *S. aureus* and *S. enteritidis* biofilms were treated with 0.5% and 1% PLA for different times. For the ultrasound (US) treatment, the mature biofilms were placed in an ultrasonic cleaning machine with 270 W and 50 kHz (Kunshan Ultrasonic, Inc., Suzhou, China). For the combined treatments (US + 0.5% PLA and US + 1% PLA), the mature biofilm cells were immersed in 0.5% and 1% PLA and treated with US for different times.

### 2.3. Biofilm Cultivation and Treatment

Approximately 10^7^ CFU (1 mL) of *S. aureus* and *S. enteritidis* cells were added into each well of Nunc™ 24-Well Flat-Bottom Plates (ThermoFisher Scientific, Kamstrupvej, Denmark). The plates were cultured statically at 37 °C for 3 days. BHI broth was changed every 24 h by removing the old medium along the walls of each well, and fresh medium was then added to maintain bacterial activity. After 3 days of cultivation, the supernatant was removed, and the biofilms were rinsed with 1 mL of 0.01 M phosphate buffer (PBS) three times. The biofilms were subjected to different antibacterial treatments, including US (270 W), 0.5% PLA, 1% PLA, and the combination of ultrasound and PLA (US + 0.5% PLA, US + 1% PLA) for 5, 10, 20, 30, and 60 min, respectively.

### 2.4. Enumeration of Biofilm Cells

A reduction in biofilm cells can directly reflect the antibacterial effect of different treatments on *S. aureus* and *S. enteritidis* biofilms. After different treatments, the water or chemicals in each well were replaced with 0.1 M PBS and kept for 10 min before aspiration. A total of 1 mL of 0.01 M PBS was then added to each well. The bacterial cells in each well were collected with a sterile cotton swab, and then diluted tenfold. The bacterial cells in the tube with an appropriate gradient were plated on BHI agar and cultured at 37 °C for 24 h before counting.

### 2.5. Polysaccharide Content in Biofilms

After different treatments, the biofilm contents in each well were collected as above. The biofilm contents from the three wells corresponding to one sample were mixed and centrifuged at 6000× *g* for 30 min at 4 °C. The soluble polysaccharide content was determined using the supernatant. The concentrated precipitate was added to a solution (0.85% NaCl, 0.22% formaldehyde) and heated for 30 min at 80 °C. The solution was further centrifuged at 12,000× *g* (4 °C, 30 min), and the supernatant was used to determine the contents of insoluble polysaccharides in the biofilms. The polysaccharide contents were quantitatively determined by the phenol–sulfuric acid method [22].

### 2.6. Confocal Laser Scanning Microscopy (CLSM) Analysis

In brief, 400 μL of approximately 10^7^ CFU/mL *S. aureus* or *S. enteritidis* was added to each well of the Nunc™Lab-Tek™ chamber slides (ThermoFisher Scientific, Kamstrupvej, Denmark). The broth was changed every 24 h to keep the bacteria active. The biofilms in the chambers were treated according to the method in Section 2.3. The biofilms were dyed and observed according to previous reports [20,21].

### 2.7. Scanning Electron Microscopy (SEM)

Microstructural changes in the bacterial cells were further observed by SEM [19,20,23]. The cultivation and antibacterial treatment of *S. aureus* or *S. enteritidis* biofilms were conducted in accordance with the method in Section 2.3. After treatment, the slides were cut according to the sample grid by using a glass knife and completely immersed in 2.5% (*v*/*v*) glutaraldehyde solution. The samples were placed at 4 °C for 12 h, dried, and placed again with 1% (*v*/*v*) osmic acid again for 90 min. After gradient dehydration, the treated slides were sprayed with gold and observed using SEM (EVO-LS10, Zeiss, Jena, Germany). The magnification was selected as ×5000 when taking pictures.

### 2.8. Release of Intracellular ATP

The mature biofilm cells of *S. aureus* and *S. enteritidis* cultured in a 24-well plate for 3 days were processed with different antibacterial treatments for 30 min according to the methodology in Section 2.3. After treatment, the supernatant was obtained by centrifugation at 10,000× *g* for 5 min. The ATP content in the supernatant was detected using the ATP detection kit (Beyotime, Shanghai, China).

### 2.9. Respiratory Chain Dehydrogenase (RCD) Determination

The activity of RCD was determined by the triphenyltetrazole chloride (TTC) method according to a previous study [24]. The mature biofilm cells of *S. aureus* and *S. enteritidis* cultured in a 24-well plate for 3 days were subjected to different antibacterial treatments for 30 min according to the methodology in Section 2.3. After treatment, the bacterial suspension of 1.0 mL was mixed with 2 mL of 0.1 mol/L glucose solution, 2 mL of 0.05 mol/L Tris-HCl buffer (pH = 8.6), and 2 mL of 1 mg/mL TTC solution. The mixture was kept in an incubator at 37 °C for 5 h and 5 mL of chloroform was added. The lower organic phase was collected into a 10 mL centrifuge tube and centrifuged at 5000× *g* (10 min, 4 °C). The absorbance at 490 nm was measured.

### 2.10. Statistical Analysis

All experiments were conducted in triplicate, of which the data were expressed as the mean ± standard deviation (SD). Statistical analysis was done through ANOVA by using SPSS software version 26.0. The significance level between groups was set at *p* < 0.05.

## 3. Results

### 3.1. Inactivation of S. aureus and S. enteritidis Biofilm Cells

Table 1 shows the effects of different treatments (ultrasound, PLA, and ultrasound combined with PLA) on inactivating the *S. aureus* bacterial cells in biofilms. The number of *S. aureus* cells in the control biofilm was approximately 9.4 log CFU/mL. The biofilm cells were not significantly reduced after treatment with single US for 30 min (*p* > 0.05) and were only reduced by 0.3 log CFU/mL when the treatment time was extended to 60 min. Hence, single US treatment was not effective to inactivate *S. aureus* biofilm cells. The treatment of 1% PLA was more effective in significantly inactivating *S. aureus* biofilm cells than the treatment of 0.5% PLA (*p* < 0.05). *S. aureus* biofilm cells were inactivated by 0.4 and 0.9 log CFU/mL after treatments with 0.5% PLA for 5 and 30 min and by 1.7 and 3.6 log CFU/mL after treatments with 1% PLA for 5 and 30 min. The combined treatment of US + 0.5% PLA for 5 and 30 min inactivated *S. aureus* biofilm cells by 1.7 and 3.4 log CFU/mL, respectively, similar to the 1% PLA treatment. These results showed that the combined treatment of ultrasound and PLA was significantly more effective than the single treatment (*p* < 0.05).

Table 2 presents the effects of different treatments on inactivating *S. enteritidis* bacterial cells in biofilms. The number of *S. enteritidis* cells in the control biofilm was approximately 8.5 log CFU/mL. *S. enteritidis* biofilm cells were inactivated by 0, 0.2, and 1.0 log CFU/mL after a single treatment of US, 0.5% PLA, and 1% PLA for 5 min and by 0.4, 2.0, and 3.1 log CFU/mL when the treatment time reached 30 min, respectively. *S. enteritidis* biofilm cells were inactivated by 4.8 and 5.0 log CFU/mL after the combined treatments of US + 0.5% PLA and US + 1% PLA for 5 min, respectively, which were significantly more effective than the single treatment (*p* < 0.05). Hence, ultrasound combined with PLA was more effective in inactivating *S. enteritidis* biofilm cells than the single treatment. The combined treatment of ultrasound and PLA significantly improved the efficiency of inactivating *S. aureus* and *S. enteritidis* biofilm cells (*p* < 0.05).

### 3.2. EPS Contents in Biofilms

During the formation of bacterial biofilms, planktonic bacteria first adhere to the solid surface and then form a polymer matrix that is rich in proteins, nucleic acids, and extracellular polysaccharides (EPS); this matrix exhibits a protective effect against adverse environmental factors. EPS is the main component of the polymer matrix. Figure 1 shows the changes in the EPS contents in *S. aureus* and *S. enteritidis* biofilms after different treatments (US, 0.5% PLA, 1% PLA, US + 0.5% PLA, and US + 1% PLA) for 30 min. As shown in Figure 1A, the soluble EPS content in the control *S. aureus* biofilm was 31.4 μg/mL. After treatment with US, 0.5% PLA, and 1% PLA, the soluble EPS content in the biofilms decreased to 23.9, 23.2, and 16.1 μg/mL, respectively. Treatments with US, 0.5% PLA, and 1% PLA reduced the soluble EPS content by 26.8%, 26.9%, and 48.7%, respectively. The combined treatment of US and PLA disrupted the EPS structure in the biofilm matrix, thereby destroying the integrity of the biofilms. Moreover, the combined treatment of US and 1% PLA decreased the soluble EPS content in the biofilms to 5.9 μg/mL. This finding indicates that the combination of ultrasound and PLA significantly reduced the insoluble EPS compared to ultrasound or PLA alone (*p* < 0.05). As shown in Figure 1B, the insoluble EPS content in the control *S. aureus* biofilm was 128.4 μg/mL. After treatment with US, 0.5% PLA, 1% PLA, US + 0.5% PLA, and US + 1% PLA, the insoluble EPS contents decreased to 108.6, 103.7, 95.5, 87.8, and 85.3 μg/mL, respectively. Treatment with 1% PLA reduced the insoluble EPS content by 25.6%, while the co-treatment of US + 0.5% PLA reduced the content by 31.6%, indicating the synergistic effect of US and PLA on removing the biofilm matrix.

Based on Figure 1C,D, the changes in the soluble and insoluble EPS contents in *S. enteritidis* biofilms after different antibacterial treatment were similar to those of the *S. aureus* biofilms. The combined treatment of US and PLA significantly decreased the soluble and insoluble EPS contents than single US or PLA treatment (*p* < 0.05). Therefore, US and PLA had synergistic effects on destroying the biofilm matrix. When US and PLA was used together, US allowed the easy entry of PLA into the biofilm cells by increasing the permeability of the cell membrane, so PLA had a better antibacterial effect.

### 3.3. CLSM Analysis

The CLSM images of *S. aureus* and *S. enteritidis* biofilms after different treatments for 30 min are shown Figure 2. The control *S. aureus* and *S. enteritidis* biofilms were green (Figure 2(A1,B1)). The *S. aureus* and *S. enteritidis* biofilms treated with US for 30 min were yellowish-green (Figure 2(A4,B4)), indicating that most bacterial cells in the biofilms remained intact. The *S. aureus* and *S. enteritidis* biofilms treated with 0.5% PLA and 1% PLA for 30 min were green, yellow, and red (Figure 2(A2,A3,B2,B3)). With increasing PLA concentrations, the number of red cells increased. The *S. aureus* and *S. enteritidis* biofilms treated with US + 0.5% PLA and US + 1% PLA were almost completely red, indicating extremely serious damage to the biofilm cells. The CLSM images indicated that the combination of ultrasound and PLA was more effective to destroy cell membranes in foodborne pathogen biofilms than the single treatments.

### 3.4. SEM Analysis

SEM was used to further analyze the effect of the combined treatment of US and PLA on the cell membrane damages of the *S. aureus* and *S. enteritidis* biofilm cells. The control bacterial cells of the *S. aureus* and *S. enteritidis* biofilms were surrounded by a large amount of EPS (Figure 3(A1,B1)). After treatment with ultrasound, 0.5% PLA, and 1% PLA for 30 min, the pathogen cells began to be exposed without the surrounding EPS (Figure 3(A2–A4,B2–B4)). Most of the *S. aureus* biofilm cells after single treatments still had cell structures (Figure 3(A4,B4)), while the *S. enteritidis* biofilm cells were seriously wrinkled (Figure 3(A2,A3,B2,B3)). The combination of ultrasound and PLA induced more obvious morphological damage to the bacterial cells (Figure 3(A5,A6,B5,B6)). The *S. aureus* biofilm cells began to shrivel (Figure 3(A5,A6)), and the *S. enteritidis* biofilm cells were thoroughly damaged and did not have intact cell structures (Figure 3(B5,B6)). Hence, ultrasound combined with PLA treatment caused more serious damage to the bacterial cells and biofilm matrix, consistent with the above results of bacterial inactivation and EPS content.

### 3.5. Release of Intracellular ATP

Extracellular ATP level is another indicator of cell damage. When the biofilm cells are damaged, ATP will be released from the cells. Figure 4 shows the changes in the extracellular ATP contents of the *S. aureus* and *S. enteritidis* biofilms after different antibacterial treatments. The contents of extracellular ATP of the control *S. aureus* and *S. enteritidis* biofilms were approximately 250 and 100 nmol/OD, respectively. After treatments with US, 0.5% PLA, 1% PLA, US + 0.5% PLA, and US + 1% PLA for 30 min, the contents of extracellular ATP in the *S. aureus* biofilm increased to approximately 500, 1250, 1600, 1750, and 2750 nmol/OD, respectively. As shown in Figure 4B, the extracellular ATP contents in the *S. enteritidis* biofilms increased to 227, 446, 689, 925, and 1480 nmol/OD after treatments with US, 0.5% PLA, 1% PLA, US + 0.5% PLA, and US + 1% PLA for 30 min, respectively. The extracellular ATP level of the biofilm treated with US + 1% PLA was significantly higher than that of the samples treated with the single US or PLA (*p* < 0.05). The combination of US and PLA was significantly more effective in causing damages to the cell structures and leakage of intracellular substances.

### 3.6. Activity of Respiratory Chain Dehydrogenase (RCD)

The inactivation of bacterial RCD activity indicates that the bacterial mechanism pathway is interrupted, which results in bacterial death [25]. As shown in Figure 5A, the OD490 value of the control *S. aureus* biofilm cells representing the activity of RCD was about 0.47. After treatment with US, the OD490 value decreased to 0.41, indicating that the activity of RCA in the US-treated biofilm decreased by 14.9% compared to the control. After treatment with 0.5% and 1% PLA, the OD490 values decreased to 0.15 and 0.11, indicating that the RCA activity in the two treated biofilm cells was decreased by 68.1% and 76.6%, respectively. The activity of RCD was more easily inactivated by the PLA than US treatment. When the combined treatment of PLA and US was used, the reduction in RCD activity was significantly greater (*p* < 0.05) than that of the single 1% PLA treatment. As shown in Figure 5B, the single US and PLA treatment also decreased the activity of RCD in the *S. enteritidis* biofilm cells to different levels. The combination of US and PLA significantly decreased the activity of RCD in the *S. enteritidis* biofilm cells compared to the single treatments (*p* < 0.05). Hence, US treatment can help PLA to enter the biofilm cells, resulting in the inactivation of RCD at high levels.

## 4. Discussion

Food safety incidents caused by food-borne pathogen biofilms have attracted wide research attention. Many different antibacterial treatments have been used to eliminate or reduce the ability of food-borne pathogens to form biofilms [26,27,28]. Li et al. [29] found that ultrasonic treatment caused multi-target inactivation of *S. aureus* and *E. coli* due to different degrees of irreversible damages to the cell wall and plasma membrane. This finding is consistent with the report of Ashokkumar [30], who found that the mechanical force generated by ultrasound through cavitation damages microorganisms, leading to a bactericidal effect. Li et al. [29] also concluded that ultrasound sterilization can destroy the integrity of cell membranes, causing intracellular esterase inactivation and inhibiting cell metabolism. The sterilization mechanism of ultrasound has been explored. However, Gao et al. [31] found that although ultrasound treatment can damage bacterial cells through physical action, the probability of inactivating the bacteria is not high, similar to our research results. In the present study, differences between the US and control groups of *S. aureus* and *S. enteritidis* were almost insignificant, and the total number of colonies was not considerably different. The comparison of SEM and CLSM images showed limited difference between the US and control groups. Compared with the PLA treatment, the US treatment only changed the distance without causing too much cell breakage. This finding verifies that the role of ultrasound in the antibacterial process is through physical action to increase the cell membrane permeability but has no significant effect on killing bacteria. Based on analysis of the bacterial counts, changes in the RCD and ATP values, and changes in the SEM and CLSM images, the changes in the combined treatment group were more significant than those in the PLA treatment alone, and PLA treatment was significantly stronger than US treatment.

PLA is often used in production as an organic acid with broad-spectrum antibacterial activity. Mu et al. [32] found that PLA can simultaneously inhibit Gram-positive and Gram-negative bacteria, including common pathogenic bacteria, such as *S. aureus* and *Salmonella*. Lavermicocca et al. [33] and Schwenninger et al. [34] reported that PLA has a good inhibitory effect on yeasts and molds. Liu et al. [35] used PLA to treat *Enterobacter cloacae* and found that it had a significant antibacterial effect; that is, 1% PLA treatment for 10 min can inactivate the biofilm by 2.8 log. In addition, Liu et al. [36] found that the combination of PLA and slightly acidic electrolyzed water was more effective to inactivate *Klebsiella* planktonic and biofilm cells than a single treatment. In the present study, for the first time, ultrasound and PLA were combined and their antibacterial effects against *S. aureus* and *Salmonella* biofilm cells were evaluated. The extracellular matrix in the biofilm contains a complex three-dimensional structure, and extracellular polysaccharides play a good protective role, leading to difficulty in removing the biofilm; as such, we innovatively adopted the combination of ultrasound and PLA to explore their joint bactericidal effect. Based on the changes in the various indicators tested, we can conclude the mechanism of ultrasound and PLA combined sterilization. Ultrasound increases the permeability of the cell membrane through cavitation and promotes the entry of PLA into bacterial cells to exert a bactericidal effect, thereby improving the sterilization ability.

Future studies should explore the combination of other sterilization methods to exert antibacterial effects. This work is the first to determine the antibacterial effect of ultrasound combined with PLA. The results provide a new and innovative antibacterial method for the food industry. PLA is often considered a green antibacterial agent owing to its natural origin; as such, the combination of US and PLA can be used to avoid the harmful effects of foodborne *S. aureus* and *S. enteritidis* in ready-to-eat meat and poultry products. At present, the cost of PLA usage is higher than the other typically used antibacterial agents due to the limitation of the large-scale industrial production of PLA. However, with the development of genetic engineering tools, PLA production will be enhanced by the fermentation of lactic acid bacteria strains. The application of PLA will be more popular due to the reduction in cost and because it satisfies the requirements for safe antibacterial agents.

## 5. Conclusions

US and PLA had a synergistic effect on inactivating the biofilm cells of *S. aureus* and *S. enteritidis*. The combined treatment significantly destroyed the cell membrane structures of the biofilm cells and led to the leakage of the internal components and inactivation of internal enzymes. In summary, US can further enhance the bactericidal effect of PLA by increasing the permeability of bacterial cells; as such, when the two methods were combined, they would have a stronger bactericidal effect than the single treatments.

## Figures and Tables

**Figure 1 foods-10-02171-f001:**
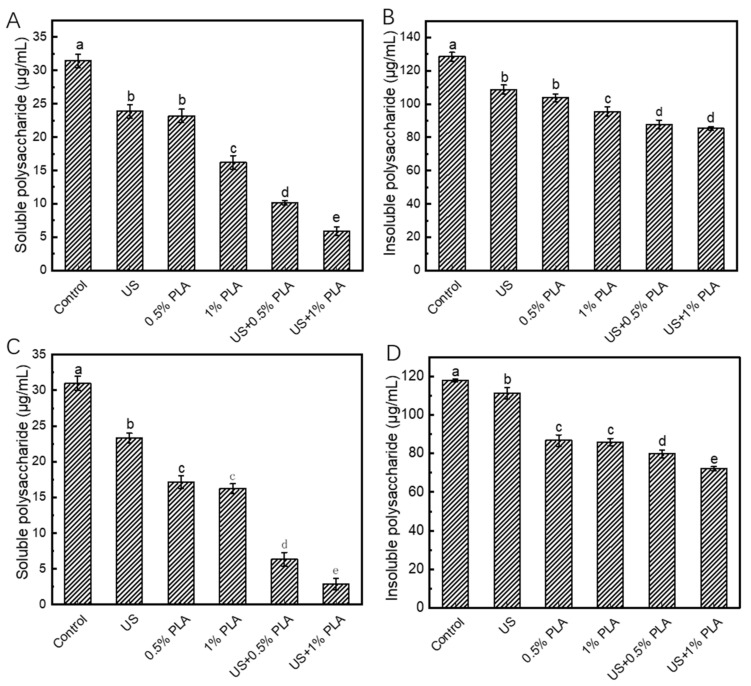
Changes in the contents of extracellular polysaccharides in *S. aureus* (**A**,**B**) and *S. enteritidis* (**C**,**D**) biofilms after different treatments. Different lower cases indicate significant difference (*p* < 0.05). US, ultrasound; PLA, phenyllactic acid.

**Figure 2 foods-10-02171-f002:**
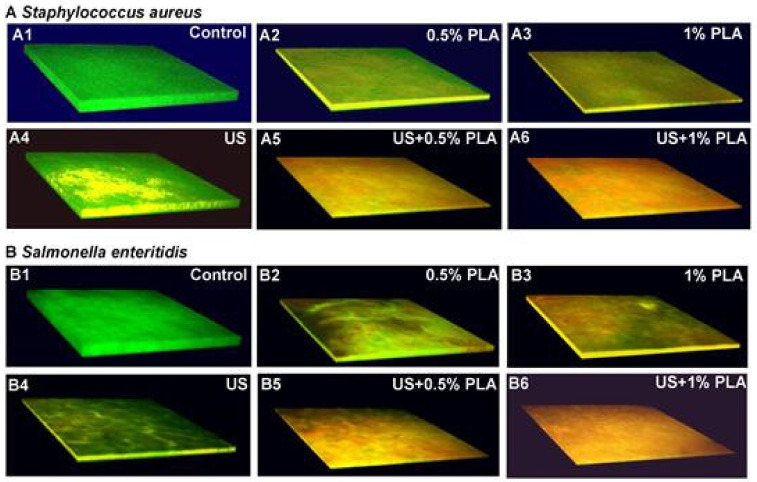
Confocal laser scanning images of *S. aureus* (**A**) and *S. enteritidis* (**B**) after the single and combined treatment of US and PLA for 30 min. US: Ultrasound; PLA: Phenyllactic acid; US + PLA: Ultrasound combined with PLA.

**Figure 3 foods-10-02171-f003:**
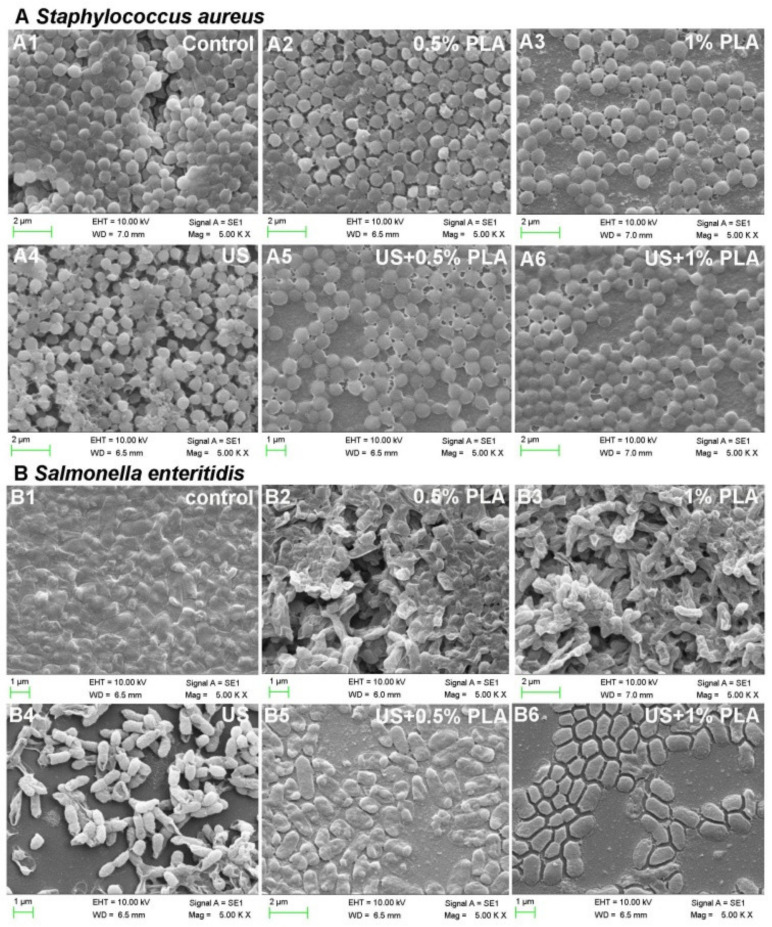
Scanning electron microscopy of *S. aureus* (**A**) and *S. enteritidis* (**B**) after the single and combined treatment of US and PLA for 30 min. US, ultrasound; PLA, phenyllactic acid.

**Figure 4 foods-10-02171-f004:**
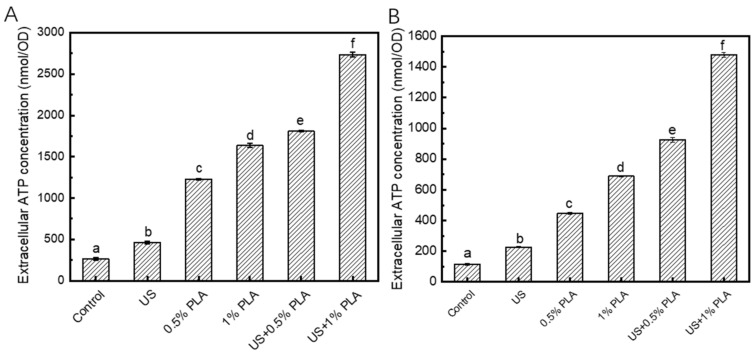
Changes in the contents of the extracellular ATP levels of *S. aureus* (**A**) and *S. enteritidis* (**B**) biofilm cells after different treatments. Different lower cases indicate significant difference (*p* < 0.05). US, ultrasound; PLA, phenyllactic acid.

**Figure 5 foods-10-02171-f005:**
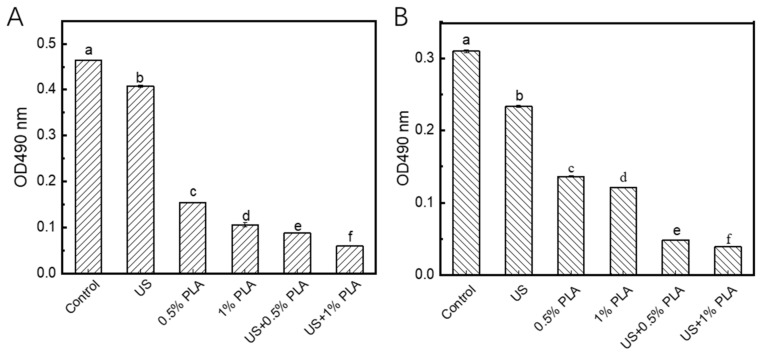
Changes in the activities of respiratory chain dehydrogenases in *S. aureus* (**A**) and *S. enteritidis* (**B**) biofilm cells. US, ultrasound; PLA, phenyllactic acid. Different letters indicate significant difference (*p* < 0.05).

**Table 1 foods-10-02171-t001:** Residual bacterial counts in *S. aureus* biofilms cells after the different treatments.

Different Treatments	*S. aureus* (log_10_CFU/mL)				
	5 min	10 min	20 min	30 min	60 min
Control	9.4 ± 0.52 Aa	9.5 ± 0.09 Aa	9.5 ± 0.03 Aa	9.5 ± 0.03 Aa	9.6 ± 0.02 Ba
US	9.3 ± 0.49 Aa	9.3 ± 0.18 Aa	9.1 ± 0.10 Bb	9.1 ± 0.24 Bab	9.3 ± 0.13 Ab
0.5% PLA	9.0 ± 0.02 Aa	8.9 ± 0.01 Ab	8.9 ± 0.09 Ac	8.6 ± 0.07 Bb	7.9 ± 0.37 Cb
1% PLA	7.7 ± 0.04 Ab	7.6 ± 0.24 Ac	6.9 ± 0.08 Bd	5.9 ± 0.55 Cc	5.8 ± 0.27 Cc
US + 0.5% PLA	7.7 ± 0.33 Ab	7.6 ± 0.37 Ac	6.8 ± 0.08 Bd	6.1 ± 0.36 Cc	5.7 ± 0.31 Dc
US + 1% PLA	6.1 ± 0.08 Bc	6.3 ± 0.05 Ad	6.2 ± 0.03 ABe	5.3 ± 0.36 Cd	5.3 ± 0.44 Cd

Note: US, ultrasound; PLA, phenyllactic acid. Different upper cases (A–D) indicate significant differences of the data in the same row (*p* < 0.05). Different lower cases (a–e) indicate significant differences of the data in the same column (*p* < 0.05).

**Table 2 foods-10-02171-t002:** Residual bacterial counts in *S. enteritidis* biofilms cells after the different treatments.

Different Treatments	*S. enteritidis* (log_10_CFU/mL)				
	5 min	10 min	20 min	30 min	60 min
Control	8.5 ± 0.06 Aa	8.6 ± 0.11 Aa	8.5 ± 0.10 Aa	8.5 ± 0.05 Aa	8.5 ± 0.01 Aa
US	8.5 ± 0.04 Aab	8.5 ± 0.21 Aa	8.4 ± 0.09 Ab	7.6 ± 0.24 Bb	7.5 ± 0.13 Bb
0.5% PLA	8.3 ± 0.04 Ab	7.7 ± 0.24 Bb	7.6 ± 0.06 Bc	6.5 ± 0.19 Cc	5.7 ± 0.08 Dc
1% PLA	7.5 ± 0.07 Ac	7.4 ± 0.03 ABc	7.3 ± 0.06 Bd	5.4 ± 0.42 Cd	5.3 ± 0.15 Cd
US + 0.5% PLA	3.7 ± 0.03 Ad	3.5 ± 0.10 Bd	3.4 ± 0.07 Be	3.2 ± 0.15 Ce	3.1 ± 0.09 Ce
US + 1% PLA	3.5 ± 0.28 Ad	3.3 ± 0.23 Bd	3.2 ± 0.04 Bf	2.9 ± 0.05 Ce	2.4 ± 0.11 Df

Note: US, ultrasound; PLA, phenyllactic acid. Different upper cases (A–D) indicate significant differences of the data in the same row (*p* < 0.05). Different lower cases (a–f) indicate significant differences of the data in the same column (*p* < 0.05).

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
