# Peer review of "Synergistic Antibiofilm Effects of Ultrasound and Phenyllactic Acid against Staphylococcus aureus and Salmonella enteritidis"

_foods, 2021, doi:10.3390/foods10092171_

Round 1

Reviewer 1 Report

The manuscript "Synergistic antibiofilm effects of ultrasound and phenyllactic
acid against Staphylococcus aureus and Salmonella enteritidis" by Jiaojiao et al., evaluates the application of ultrasound and phenyllactic acid in combination for the treatment of biofilms formed by Staphylococcus aureus and Salmonella enteritidis

The major concern about the manuscript is the experimental design and the novelity of the work. 

Why the authors have not included antimicrobial activity as an evaluation parameter and then check the antibiofilm activity. Is it really just the antibiofilm activity or an improved antimicrobial activity.

Section 2.3: This section is not clearly demonstrated that when the treatment was performed. 

Was the treatment was performed directly on the biofilm? if so it is also advised to treat the cells and let it grow to understand if it also disturb the biofilm formation.

Throughout the manuscript, it is advised to check the gramatical errors and typos.

One such example is to write genus sp names in Italics throughout.

 "This finding verifies that the role of ultrasound in the antibacterial process is through physical action to increase the cell membrane permeability but has no significant effect on killing bacteria." Is it so?

In order to justify this statement antimicrobial activity should be included in the study together with membrane permeability assay.

Author Response

Review 1

The manuscript "Synergistic antibiofilm effects of ultrasound and phenyllactic acid against Staphylococcus aureus and Salmonella enteritidis" by Jiaojiao et al., evaluates the application of ultrasound and phenyllactic acid in combination for the treatment of biofilms formed by Staphylococcus aureus and Salmonella enteritidis.

The major concern about the manuscript is the experimental design and the novelity of the work.

Why the authors have not included antimicrobial activity as an evaluation parameter and then check the antibiofilm activity. Is it really just the antibiofilm activity or an improved antimicrobial activity?

Response: Thanks for the suggestion. We also studied the effect of combination of ultrasound and phenyllactic acid to inactivate S. aureus and S. enteritidis planktonic cells. The planktonic cells were more effective to be inactivated than the biofilm cells. This paper is focused the inactivation of biofilm cells, so we only show the antibiofilm data.

Section 2.3: This section is not clearly demonstrated that when the treatment was performed.

Response: The point has been taken in the revision (Line 83-87).

Was the treatment was performed directly on the biofilm? if so it is also advised to treat the cells and let it grow to understand if it also disturb the biofilm formation.

Response: Yes, we want to analyze the effect of different treatments to inactivate bacterial cells in mature biofilms, so the treatment was performed directly on the biofilms.

Throughout the manuscript, it is advised to check the grammatical errors and typos.

Response: The English language was further revised by an English editing company, and the grammatical errors and typos were also revised.

One such example is to write genus sp names in Italics throughout.

Response: The genus sp names were revised (Line 83-87).

"This finding verifies that the role of ultrasound in the antibacterial process is through physical action to increase the cell membrane permeability but has no significant effect on killing bacteria." Is it so? In order to justify this statement antimicrobial activity should be included in the study together with membrane permeability assay.

Response: The description of this sentence is not appropriate, and it was revised (Line 296-298).

Reviewer 2 Report

This manuscript aimed to evaluate the synergistic antibiofilm effects of ultrasound and phenyllactic acid against Staphylococcus aureus and Salmonella enteritidis. It is an interesting subject; however several points need to be addressed.

Specific comments:

  1. 32-34: Reference 2 does not support the respective comment. Authors need to find an appropriate one.
  2. 51: Authors need to give information related to the antibacterial effect mechanism of PLA.
  3. 57: Authors need to refer to previous studies related to the combination of ultrasound and antimicrobial compounds to inactivate the targeted foodborne pathogens. Some of this information appears in the discussion section (L. 317-323) but needs to be moved to the introduction section. Authors need also to highlight the gap that their approach would cover (e.g. more cost effective and safe approach?).
  4. 67: Where were cells transferred? It is not clear.
  5. 70-76: In general, the experimental design followed the One-Factor-Per-Time approach, which is not the best choice to study interaction effects of independent parameters. Authors could have chosen the Response Surface Methodology and the Central Composite Design (CCD). How were the treatment parameter levels chosen in this study? Authors need to explain this issue. Extra experimentation in the frame of CCD is suggested.
  6. 80: Please change the word "extracting" to "removing".
  7. 87: What authors mean with the phrase "removal effect"?
  8. 90: Authors need to improve the phrase “Each well was then added with 0.01 M PBS (1 mL)”.
  9. 110: Change dash to comma through text.
  10. 111: The antibacterial treatment is not described in 2.4 but in 2.3.
  11. 119: This information should be removed to the respective part of the R&D section
  12. In Tables 1 and 2 authors should delete the “at min” from the first row and add a new row entitled "Treatment time (min)". Also, the upper cases (A-G) are not shown in the tables.
  13. The microbial species should be written in italics through text.
  14. In the discussion section, authors should discuss in more details the advantages and applicability of their approach to food manufacturing in terms of feasibility (e.g. in which type of foods the combined effect of US and PLA could be applied, is it cost effective compared with other antimicrobial compounds?) and safety issues (toxicity level of antimicrobial agent etc).

Author Response

Review 2

This manuscript aimed to evaluate the synergistic antibiofilm effects of ultrasound and phenyllactic acid against Staphylococcus aureus and Salmonella enteritidis. It is an interesting subject; however several points need to be addressed.

Specific comments:

32-34: Reference 2 does not support the respective comment. Authors need to find an appropriate one.

Response: Reference 2 was revised to another appropriate one.

51: Authors need to give information related to the antibacterial effect mechanism of PLA.

57: Authors need to refer to previous studies related to the combination of ultrasound and antimicrobial compounds to inactivate the targeted foodborne pathogens. Some of this information appears in the discussion section (L. 317-323) but needs to be moved to the introduction section. Authors need also to highlight the gap that their approach would cover (e.g. more cost effective and safe approach?).

Response: The previous studies related to the combination of ultrasound and antimicrobial compounds to inactivate the targeted foodborne pathogens shown in the discussion section was revised to the introduction section (Line 53-57). The cost effective and safe approach was added in the discussion section.

67: Where were cells transferred? It is not clear.

Response: The point has been taken in the revision (Line 67).

70-76: In general, the experimental design followed the One-Factor-Per-Time approach, which is not the best choice to study interaction effects of independent parameters. Authors could have chosen the Response Surface Methodology and the Central Composite Design (CCD). How were the treatment parameter levels chosen in this study? Authors need to explain this issue. Extra experimentation in the frame of CCD is suggested.

Response: Thank you for your suggestion. Generally speaking, the Response Surface Methodology and the Central Composite Design (CCD) were used to get an optimal treatment condition. In our study, we want to show the changes of bacterial counts in biofilms under different treatments for different time. These data were used to show the synergistic antibacterial effect when the US and PLA were used together to inactivate biofilm cells according to our previous paper.

Fang Liu, Chun Tang, Debao Wang, Zhilan Sun, Lihui Du, Daoying Wang, 2021, he synergistic effects of phenyllactic acid and slightly acid electrolyzed water to effectively inactivate Klebsiella oxytoca planktonic and biofilm cells, Food Control, 125, 107804.

80: Please change the word "extracting" to "removing".

Response: The point has been taken in the revision (Line 67).

87: What authors mean with the phrase "removal effect"?

Response: The removal effect was revised to “antibacterial effect”.

90: Authors need to improve the phrase “Each well was then added with 0.01 M PBS (1 mL)”.

Response: This sentence was revised to “One ml of 0.01 M PBS was then added to each well.”

110: Change dash to comma through text.

Response: These points were revised.

111: The antibacterial treatment is not described in 2.4 but in 2.3.

Response: The point has been taken in the revision (Line 113).

119: This information should be removed to the respective part of the R&D section.

Response: This information had removed to part 3.5 of the R&D section (Line 243-244).

In Tables 1 and 2 authors should delete the “at min” from the first row and add a new row entitled "Treatment time (min)". Also, the upper cases (A-G) are not shown in the tables.

Response: These points were revised in Tables 1 and 2.

The microbial species should be written in italics through text.

Response: These points were revised.

In the discussion section, authors should discuss in more details the advantages and applicability of their approach to food manufacturing in terms of feasibility (e.g. in which type of foods the combined effect of US and PLA could be applied, is it cost effective compared with other antimicrobial compounds?) and safety issues (toxicity level of antimicrobial agent etc).

Response: This was added in the discussion section of the revision (Line 331-339).

Reviewer 3 Report

Comments and Suggestions for Authors

This paper deals synergistic antibiofilm effects of ultrasound and phenyllactic acid 
against Staphylococcus aureus and Salmonella enteritidis receiving a lot of attention from researchers around the world in recent years. Therefore, the method to be used must be very rigorous and precise and should follow the guideline with the necessary modifications Methods should be described in detail, controls should be used (a conventional drug, a bacteria test control, etc.); it is necessary to add breakpoint of methods

The paper may be interesting but the methodology is accurate. I think the Authors should to consider a substantially revision of the manuscript and to carefully ensure that the English language is correct as there are many issues with English grammar.

Please it is reccomendend to inserti  in the text and in the bibliography the paper
Ultrasound affects minimal inhibitory concentration of ampicillin against methicillin resistant Staphylococcus aureus USA300 (2019)

Author Response

Review 3

This paper deals synergistic antibiofilm effects of ultrasound and phenyllactic acid against Staphylococcus aureus and Salmonella enteritidis receiving a lot of attention from researchers around the world in recent years. Therefore, the method to be used must be very rigorous and precise and should follow the guideline with the necessary modifications Methods should be described in detail, controls should be used (a conventional drug, a bacteria test control, etc.); it is necessary to add breakpoint of methods

Response: The method was revised and some information was added.

The paper may be interesting but the methodology is accurate. I think the Authors should to consider a substantially revision of the manuscript and to carefully ensure that the English language is correct as there are many issues with English grammar.

Response: The English language was revised by an English editing company.

Please it is reccomendend to insert in the text and in the bibliography the paper

Ultrasound affects minimal inhibitory concentration of ampicillin against methicillin resistant Staphylococcus aureus USA300 (2019)

Response: This reference was inserted in the instruction part of the revision.

Round 2

Reviewer 1 Report

The manuscript has improved substantially after the careful revision by the authors.